# A Novel Postbiotic Product Based on *Weissella cibaria* for Enhancing Disease Resistance in Rainbow Trout: Aquaculture Application

**DOI:** 10.3390/ani14050744

**Published:** 2024-02-27

**Authors:** Mario Quintanilla-Pineda, Francisco C. Ibañez, Chajira Garrote-Achou, Florencio Marzo

**Affiliations:** 1Laboratorio de Fisiología y Nutrición Animal, Universidad Pública de Navarra, 31006 Pamplona, Spain; pi@unavarra.es; 2PENTABIOL SL, 31191 Pamplona, Spain; chajira@pentabiol.es

**Keywords:** immunomodulation, salmonids, acid-lactic bacteria, microbiota modulation, growth performance, Red Mouth Disease, yersiniosis

## Abstract

**Simple Summary:**

Antibiotics have become a threat to aquaculture due to their negative impact on aquatic environments, fish, and consumers. To contribute to the research and creation of novel tools that are able to treat and control disease in fish and substitute the use of antibiotics in the aquaculture industry, this study investigates the impact of postbiotics added through feed on health parameters and disease resistance in rainbow trout. After 30 days of supplementation with postbiotics, an improvement in survival rate when facing an experimental infection with the causative agent of Red Mouth Disease was observed in rainbow trout. This research illustrates, for the first time, the positive effect of postbiotic supplementation on rainbow trout survival when challenged with Red Mouth Disease infection; in summary, this proves the suitability of the postbiotic product for the aquaculture industry and its relevance regarding the current efforts to reduce and replace the utilization of antibiotics in animal husbandry.

**Abstract:**

Postbiotics are innovative tools in animal husbandry, providing eco-friendly solutions for disease management within the industry. In this study, a new postbiotic product was evaluated for its impact on the health of rainbow trout (*Oncorhynchus mykiss*). In vivo studies were conducted to assess the safety of the *Weissella cibaria* strains used in postbiotic production. Additionally, this study evaluated the impact of diet supplementation with 0.50% postbiotics on growth performance during a 30-day feeding trial; the gut microbial communities, immunomodulation, and protection against *Yersinia ruckeri* infection were evaluated. The strains did not harm the animals during the 20-day observation period. Furthermore, the effect of postbiotics on growth performance was not significant (*p* < 0.05). The treated group showed a significant increase in acid-lactic bacteria on the 30th day of the feeding trial, with counts of 3.42 ± 0.21 log CFU/mL. Additionally, there was an up-regulation of the pro-inflammatory cytokine *IL-1β* in head kidney samples after 48 h of feed supplementation, whereas cytokines *IL-10*, *IL-8, INF-γ*, and *TNF-α* were down-regulated. The findings indicate that rainbow trout fed with postbiotics saw an improvement in their survival rate against *Y. ruckeri*, with a 20.66% survival improvement in the treated group. This study proves that incorporating postbiotics from two strains of *W. cibaria* previously isolated from rainbow trout into the diet of fish has immunomodulatory effects, enhances intestinal microbial composition, and improves fish resistance against *Y. ruckeri.*

## 1. Introduction

Rainbow trout (*Oncorhynchus mykiss)* was the primary fish species cultured in the European Union in 2020, with a significant 6.3% increase in production compared with the previous year [1]. Despite its growth, rainbow trout production has faced significant challenges, particularly in dealing with bacterial diseases. These include *Flavobacterium* spp., *Aeromonas* spp., and *Renibacterium* spp., as well as the causative agent of Red Mouth Disease (RMD), *Yersinia ruckeri* [2,3]. *Y. ruckeri* is a Gram-negative rod with rounded ends that does not form spores or capsules, but does have flagella [4]. *Y. ruckeri* is a pathogenic bacterium that is distributed worldwide, both through human-mediated intervention and intrinsic factors [5,6,7]. Although it is commonly associated with salmonids, it can also affect other fish species such as channel catfish (*Ictalurus punctatus*), sturgeon (*Acipenser baeri*), and goldfish (*Carassius auratus*) [8].

The prevalence of RMD has led to the use of antibiotics, which has contributed to the emergence of antibiotic-resistant bacteria. This has far-reaching consequences, affecting not only the aquatic environment, but also consumers [9]. In response, the scientific community has focused on developing new environmentally friendly solutions that provide antibacterial treatment and promote animal health [10]. Postbiotics have emerged as a safer and more stable alternative to probiotics. This is because they do not require the maintenance of live microorganisms during storage and industrial processing. Additionally, there is no risk of bacteremia associated with the administration of microorganisms that can replicate [11].

In recent years, postbiotics have demonstrated their value in the aquaculture industry as antimicrobials, intestinal microbiota modulators, immunomodulators, and protectors from pathogens [10]. Studies have shown that the administration of the cellular components of the probiotics *Aeromonas sobria* and *Bacillus subtilis* has the potential to protect rainbow trout from experimental infection with *Y. ruckeri* [12].

Recent studies have shown that *W. cibaria* strains isolated from fish have antimicrobial potential, effectively inhibiting the pathogens *Aeromonas salmonicida* subsp. *Salmonicida,* and *Y. ruckeri* [13]. Furthermore, the *Weissella* genus has emerged as a promising candidate for probiotic use due to its ability to enhance intestinal microbiota, improve growth performance, and improve haemato-immunological parameters [14]. The beneficial properties of *Weissella*, specifically of *W. cibaria,* are attributed to its ability to produce oligosaccharides and polysaccharides that can be used as prebiotics; it also produces protein-like molecules, metabolites, and bacteriocins with antimicrobial activity, which effectively inhibit the growth of pathogens [15,16]. In the field of aquaculture, the *W. cibaria* species has shown promise as a probiotic by increasing survival rates in crucian carp after *Aeromonas veronii* infection [17]. This makes the species a viable alternative to antibiotics in the aquaculture industry.

In this scenario, a postbiotic product that was previously assessed in vitro for its potential application in fish aquaculture [13,16] will be studied. The aim is to better understand its potential as a substitute for antibiotics and its ability to improve health parameters in rainbow trout. Therefore, the aim of this research is to test the hypothesis that supplementing fish feed with postbiotics produced by *W. cibiaria* can enhance the survival of rainbow trout when facing *Y. ruckeri* infection, primarily by acting as an immunomodulator and fostering improvements in the intestinal microbial communities. The objectives of this work were (i) to evaluate the biosafety of the strains used to obtain the postbiotic product; (ii) to analyze the changes in intestinal bacterial counts, immune response biomarkers, and final weight over a 30-day feeding trial with postbiotic-supplemented feed; and (iii) to determine the impact of the postbiotic product on disease resistance against *Y. ruckeri.*

## 2. Materials and Methods

### 2.1. Bacteria Strains

Two bacteria strains isolated from a fish farm in Navarre, Spain, codified as CECT 30731 and CECT 30732, belonging to *Weissella cibaria* species and isolated from the skin and gut mucus, respectively, of rainbow trout were used individually [13]. The bacteria were incubated in Man Rogosa Sharp agar and broth (MRS-A and MRS-B, Laboratorios Conda S.A., Madrid, Spain) at 30 ± 0.20 °C with 150 rpm agitation in aerobic conditions for 24 h. Afterward, cultures were washed twice by centrifugation at 2400× *g* for 10 min; the pellet was then recovered and resuspended in phosphate-buffered saline solution (PBS, Fisher BioReagents, Geel, Belgium), and the concentration was adjusted at 10^7^ CFU/g for oral administration and at 10^6^ CFU/mL for intraperitoneal administration.

### 2.2. Experimental Diets

Postbiotics were produced as previously described [16] by coculturing the strains CECT 30731 and CECT 30732 for 72 h at 30 °C in constant agitation at 150 rpm. The broth culture was then inactivated with heat at 80 °C for 1 h. The inactivation was confirmed by spreading a sample in an MRS-A plate to then be incubated for 48 h in aerobic conditions; the absence of growth ensures complete inactivation. The postbiotic product was then mixed, as described by Cabello-Olmo et al. [18], with oat bran, alfalfa meal, and non-bitter beer yeast, obtained directly from the manufacturer (PENTABIOL SL, Noain, Spain; https://www.pentabiol.es/) (accessed on 3 July 2023). The fermented product was air-dried at 80 °C for 4 h to reduce the moisture content by up to 6.40%. Then, it was micronized and mixed at a 0.50% concentration with the basal diet of commercial feed T-1.5 Nutra MP (Skretting España SA, Burgos, Spain), composed of fish meal, fish oil, wheat, soy protein concentrate, soybean meal, vitamin A, vitamin D_3_, monohydrated sulfate, potassium iodide, cobalt, copper sulfate pentahydrate, manganese sulfate monohydrate, zinc sulfate monohydrate, ammonium molybdite, sodium selenite, butylated hydroxytoluene, and butylated hydroxyanisole (https://www.skretting.com/) (accessed on 3 July 2023). The feed was stored at 4 °C.

### 2.3. Biosafety Evaluation of Postbiotic Products

Oral and intraperitoneal administration of the strains CECT 30731 and CECT 30732 was performed to evaluate their biosafety, as described by Medina et al. [19]. For this, a total of 90 healthy rainbow trout of 30.00 ± 5.50 g were obtained from Truchas de Leitza SL (Leitza, Spain) and transported via a land route to the laboratory of PENTABIOL SL located in Noain (Spain). Fish were acclimatized for two weeks in 1200 L fiberglass tanks, at 18.3 ± 0.6 °C, with constant aeration, 30% daily water replacements, and 12:12 photoperiod. Before the experimental trial was executed, the health status was microbiologically evaluated by aseptically taking samples of the spleen, head kidney, and brain, which were then spread in triplicate on Plate Count Agar (PCA, Laboratorios Conda S.A, Madrid, Spain). Plates were incubated at 22 °C for 72 h in aerobic conditions. The absence of growth confirmed the good health status of the animal batch. After the acclimatization period, animals were randomly distributed in tanks of 100 L to form six groups of nine fish each, including the following groups: a negative control for the oral challenge (fed with commercial feed without bacterial strains), a negative control for the intraperitoneal challenge (intraperitoneally injected with PBS), a group challenged with CECT 30731 strain via oral administration, a group challenged with CECT 30732 via oral administration, a group challenged with CECT 30731 strain via intraperitoneal administration, and a group challenged with CECT 30732 strain via intraperitoneal administration. Each challenge was performed in duplicate.

To perform oral administration, commercial feed T-1.5 Nutra MP was sprayed with the bacterial suspensions at 10^7^ CFU/g. For intraperitoneal administration, fish were anesthetized with 100 mg/L of tricaine methanesulfonate solution (MS 222, Pharmaq Ltd., Fordingbridge, UK) to then be intraperitoneally injected with 0.1 mL of the bacterial suspensions at 10^6^ CFU/mL. Fish were inoculated with 0.1 mL of sterile PBS and fed with commercial feed without the strains as a control.

Fish were observed daily to detect changes in behavior or mortality, or the emergence of disease symptoms, for 20 days. Afterward, fish were sacrificed with an overdose of 200 mg/L of MS 222 according to the recommendations of Directive 2010/63/EU (https://eur-lex.europa.eu/legal-content/ES/TXT/?uri=celex%3A32010L0063) (Accessed on 3 July 2023) of the European Parliament. Microbiological analysis of the head kidney and spleen was conducted as previously described to determine the presence of the inoculated strains.

### 2.4. Experimental Design

A 30-day feeding trial using the postbiotic product described above was executed to evaluate its effect on juvenile rainbow trout. For this, a total of 182 pathogen-free rainbow trout of 9.37 ± 2.20 g were obtained from Truchas de Leitza SL and transported to the laboratory. Following an acclimatization period of 15 days, fish were separated into two randomly assigned groups, in 1200 L fiberglass tanks, at 14.6 ± 0.8 °C, constant aeration, 30% daily water replacements, and 12:12 photoperiod. One group was fed as a control with unsupplemented commercial feed. The other group was treated with commercial feed supplemented at 0.5% with the postbiotic product. All fish were fed daily at 2.0% of their biomass. At the end of the 30-day feeding trial, animals were anesthetized with 100 mg/L of MS 222 and individually weighted.

### 2.5. Sample Collection

Sampling on days 7, 15, and 30 of the feeding trial to perform microbiological analyses on intestinal mucus to determine alteration in acid-lactic bacteria and total aerobic bacteria counts was conducted as described by Standenm et al. [20] by taking samples of the mid-section gut mucus from 3 randomly selected fish from each group at each sampling point. Head kidney samples were taken to evaluate cytokine gene expression on days 1, 2, 3, 4, 7, 15, and 30 of the feeding trial. To achieve this, 3 fish from each group at each sampling point were randomly selected and sacrificed; the internal organs were carefully removed to expose the head kidney which was subsequently removed and transferred to an Eppendorf of 1.5 mL. Samples were then preserved in 0.5 mL of RNAlater^TM^ Soln. (Invitrogen by Thermo Fisher Scientific, Vilnius, Lithuania), and stored at −80 °C until processing.

### 2.6. Changes in Intestinal Microbiota

Microbiological analyses were performed as previously described by Adel et al. [21]. Briefly, the collected mucus was resuspended in PBS to obtain a 1:10 solution that was vigorously vortexed for 5 min prior to being serially diluted tenfold. Trypticase Soy Agar (TSA, Laboratorios Conda S.A, Madrid, Spain) was used to estimate total viable aerobic bacterial counts and MRS-A was used for viable acid-lactic bacteria counting. After dilution, 0.1 mL was spread over the surfaces of plates and incubated at 22 °C for 72 h in aerobic conditions. The procedure was carried out in duplicate using 3 samples per group, and results were expressed as Log CFU/mL.

### 2.7. Cytokine Gene Expression

#### 2.7.1. RNA Isolation

Total RNA was extracted from samples of head kidney tissues using Maxwell^®^ RSC SimplyRNA Tissue Kit (Promega Corporation, Madison, WI, USA), according to the manufacturer’s instructions. Approximately 30 mg of tissues was homogenized with a rotating sterile pestle using a Homogenizer Unit for Microtube dDBiolab (SLU, Barcelona, Spain) in Eppendorf tubes containing 350 µL of Lysis buffer and 40 µL of Proteinase K solution. Samples were then dry-heated for 20 min at 70 °C, to then be transferred to the respective well on the MaxWell^®^ Cartridge (Promega Corporation, Madison, WI, USA). Next, 5 µL of DNase solution was added to the fourth well of the cartridge as indicated for the elimination of genomic DNA from samples. The cartridge was then placed on the tray of the Maxwell^®^ RSC Instrument (Promega Corporation, Fitchburg, MA, USA). Finally, one elution tube of 0.5 mL per cartridge with 50 µL of nuclease-free water was placed in the tray at the end of each row. The SimplyRNA Tissue processing method was selected in the Maxprep^TM^ software V1.0.1 (Maxprep^TM^ Liquid Handler Method). After RNA extraction, samples were immediately stored at −80 °C until use, and 2 µL of each sample was used to quantify RNA concentration using a SimpliNano spectrophotometer (GE Healthcare Bio-Sciences AB, Buckinghamshire, UK).

#### 2.7.2. cDNA Preparation: RNA Reverse Transcription

The transcription of total mRNA into cDNA was performed with the random primers method using BioTools Reverse transcription KIT (BioTools Biotechnological & Medical Laboratories SA, Madrid, Spain). For this step, a 30 µL reaction of a premix (2× buffer for the mRNA transcription) was prepared by adding the following reagents: 1.5 µL of BioTools enzyme (100 units/mL), 1.5 µL of dNTP mix (25 mM), 1.5 µL of random primer (9 mM), 0.25 µL of MgCl_2_ (10 mM), 0.25 µL of Dtt, 1,4-Ditiotreitol (10 mM), 3 µL of enzyme buffer (10×), 7 µL of RNAses-free water, and 15 µL of RNA extracted sample.

After the total RNA from each sample was obtained, 15 µL of the extracted RNA was added to 15 µL of reverse transcription solution mixture (2×): The reaction mixtures were then run in the thermocycler according to the following conditions: first-strand cDNA synthesis at 42 °C for 45 min and reverse-transcriptase inactivation at 70 °C for 5 min.

#### 2.7.3. Real-Time PCR (qPCR)

Real-Time PCRs were performed using the CFX Connect^TM^ Optics Module, with Bio-Rad CFX Maestro 1.1 software (Bio-Rad Laboratories, Inc., Hercules, CA, USA), using the specific primers listed in Table 1. Reaction mixtures were prepared for 20 µL total volume reaction as follows: 10 µL of 2× SYBR green PCR Mastermix Gotag-Promega (Promega Corporation, Madison, WI, USA), 0.4 µL of Forward and Reverse primers at 5 µM final concentration each, 7.7 µL of RNases-free water and 1.5 µL of cDNA from each sample. Nuclease-free water was used as negative control (NTC), and all determinations were run in triplicate. Cycling conditions for qPCR reactions are listed in Table 2.

### 2.8. Experimental Infection

For the challenge, rainbow trout with an average body weight of 17.98 ± 4.26 g from the 30-day feeding trial were used. These fish were transferred to a recirculating system of 170 L at 15.2 ± 0.81 °C, with constant aeration, 30% daily water replacements, and a 12:12 photoperiod. The challenges were carried out in duplicate with 28 fish per tank. Groups were settled as follows: two tanks of fish feed with supplemented diet infected with the pathogen, two tanks as positive control infected with the pathogen and feed with control diets, and one tank as a non-infected negative control which were fed with a control diet.

A strain of *Yersinia ruckeri* isolated in 2020 from a disease outbreak in a rainbow trout farm located in Brescia (Istituto zooprofilattico Sperimentale della Lombardia e dell’Emilia Romagna, Italy) was used. Pathogen identification was confirmed by qPCR. Briefly, the strain was cultured in Trypticase Soy Broth (TSB, Laboratorios Conda S.A., Madrid, Spain) for 24 h at 22 °C in aerobic conditions and constant agitation at 150 rpm. After incubation, cells were washed by transferring aliquots of 15 mL of the broth culture to Falcon tubes to then centrifugate at 2400× *g* for 10 min. The supernatants were discarded and the pellets were resuspended in PBS; the procedure was executed twice. The final suspension of bacteria was diluted tenfold to obtain a 9.90 × 10^7^ CFU/mL concentration and transferred to sterile syringes of 1 mL, which were immediately stored in refrigeration at 4 °C prior to use. Bacterial concentration was confirmed by microdilution of a 1 mL sample and incubating it in TSA for 24 h at 22 °C for colony-forming unity counting. Fish were then anesthetized with MS 222 as described before and inoculated intraperitoneally with 0.1 mL of the bacteria suspension. Sterile PBS was used to inoculate the negative control group. Fish were monitored daily for 17 days after exposure to *Y. ruckeri*, and each moribund or dead fish was necropsied to microbiologically confirm the cause of death.

### 2.9. Ethics Statement

This study was approved by the Director of Livestock Service of the Navarre Government Animal Welfare Section. All animal experiments were conducted according to the ethical guidelines approved by the Navarre Government, Resolution number 621E/2023, project number 2307.

### 2.10. Statistical Analyses

Differences in fish growth data collected at the end of the experiment were analyzed using an unpaired two-tailed Student’s *t*-test. Differences in microbiota and relative gene expression were statistically analyzed by one-way analyses of variance (ANOVA), followed by the least significant difference test (LSD) in cases where the values presented significant differences. Cumulative survival was analyzed by the Kaplan–Meier method and significance was determined using the chi-squared by log rank test. The level of significance was set at *p* < 0.05. All statistical analyses were performed using InfoStat version 2017 (Infostat group, Cordoba, Argentina).

## 3. Results

### 3.1. Biosafety Evaluation

After 20 days of observation of groups challenged by intraperitoneal and oral inoculation with live cells of strains CECT 30731 and CECT 30732, no mortality, and neither change in behavior nor external lesions related to bacterial infection were detected. No internal lesions were observed during necropsies at the end of the trial and the absence of bacteria was confirmed microbiologically from head kidney and spleen samples.

### 3.2. Changes in Growth and Intestinal Microbiota

After 30 days of feeding, no mortality nor significant difference (*p* < 0.05) in final weight was observed between the group treated with the 0.50% postbiotic supplemented diet (17.95 ± 3.80 g) and the control fed the unsupplemented diet (18.85 ± 4.89 g) (Figure 1).

Results of changes in intestinal microbiota following feeding with postbiotic supplementation are presented in Table 3. Unlike acid-lactic bacteria on day 30 of supplementation, there were no significant differences in either aerobic or acid-lactic bacteria counts at any of the earlier sampling points.

### 3.3. Effect on Cytokine Gene Expression

The relative fold changes at different times of the supplementation period in the expression of anti-inflammatory cytokine *IL-10,* pro-inflammatory cytokines *TNF-α, IL-1β, IL-8,* and antiviral cytokine *INF-γ* on head kidney samples of rainbow trout are presented in Figure 2.

No significant (*p* < 0.05) changes in any of the studied cytokines were observed among groups after 24 h of supplementation. Significant (*p* < 0.05) up-regulation of *IL-1β* was observed at 48 h in the treated group, whereas a significant (*p* < 0.05) down-regulation of *INF-γ* was observed at the same sampling point (Figure 2). At 72 h of the feeding trial, a significant (*p* < 0.05) down-regulation of all studied cytokines was expressed, except for *IL-1β*, which showed no significant differences compared with the control group. On day 4 of the feeding trial, only cytokines *IL-10* and *TNF-α* showed significant (*p* < 0.05) down-regulation compared with the control. No significant (*p* < 0.05) difference in any of the studied cytokines was observed between groups on days 7 and 15 of the feeding trial. Only the *IL-10* showed a significant (*p* < 0.05) down-regulation on day 30.

### 3.4. Effect on Disease Resistance to Y. ruckeri Challenge

Postbiotic administration through feed at 0.50% concentration had a significant effect (*p* < 0.05) on survival following the 17 days of observation after *Y. ruckeri* intraperitoneal infection (Figure 3). In particular, a survival rate of 64.25% was registered in the group fed with the postbiotic-supplemented diet, compared with the 43.59% survival of the group fed unsupplemented diet. Both groups showed a significant difference (*p* < 0.05) from the negative control group which was inoculated with PBS and received no dietary supplementation.

## 4. Discussion

This study utilized two strains of *W. cibaria* that have previously exhibited favorable characteristics for the development of a novel postbiotic product [13,16] to investigate potential beneficial effects in rainbow trout. To qualify as a postbiotic, one of the main criteria is to assess the safety of the preparation in the target host for its intended use [11]. To achieve this, oral and intraperitoneal administration of the two *W. cibaria* strains was performed. The results showed no mortality or complication in the inoculated fish, and the final microbiological verification confirmed the absence of the strains in internal organs. This statement indicates that the bacteria utilized in the production of the postbiotic product are not harmful to the host. This is of importance since disease outbreaks caused by other species belonging to the genus *Weissella* have been reported worldwide [23]. Additionally, no other studies regarding the use of *W. cibaria* as a postbiotic in rainbow trout have been found. Therefore, the evidence presented here could become of great importance for future research of the genus and its applications as a safe feed additive in aquaculture rearing.

Several studies have focused on the effect that postbiotic products may have on growth performance in aquaculture. For instance, studies have shown a positive effect on growth after dietary administration of postbiotics in white shrimp (*Panaeus vannamei*) and sturgeon (*Acipenser baerii x Acipenser schrenckii*) [24,25]. However, this research did not find that a significant difference (*p* < 0.05) between the treated and control group on weight gain was observed after 30 days of the feeding trial; other researchers have reported similar scenarios where no influence on growth was observed after postbiotic feed supplementation of common carp (*Cyprinus carpio*) [26,27].

The beneficial microorganism groups, such as acid-lactic bacteria in the fish gut, are not considered dominant in the intestinal microbiota [28]. However, these groups can be enhanced by adding functional ingredients or food additives to the diet. The main purpose of this enhancement is to promote the proliferation of beneficial bacteria and improve the health status of the fish [29]. In this context, researchers have studied positive effects of postbiotics on gastrointestinal microbial communities. Promising results have been found in which the so-called beneficial microbiota were modulated, improving abundance and diversity through the addition of postbiotic treatment [24,25,26]. The findings of the present research agree with these researchers’ results since a significant difference was observed in acid-lactic bacteria counts after 30 days of postbiotic supplementation. The observed colonization could become an improvement in the intestinal environment, contributing to the competition for adhesion sites and the exclusion of pathogenic bacteria on the gastrointestinal tract of the fish [30].

This study utilized different cytokines as biomarkers to evaluate the immunomodulatory effects of a postbiotic in rainbow trout. These included the anti-inflammatory cytokine *IL-10*, as well as the pro-inflammatory cytokines *TNF-α, IL-1β*, and *IL-8*, and the antiviral cytokine *INF-*γ. However, only the pro-inflammatory cytokine *IL-1β* was up-regulated in head kidney samples after 48 h of feed supplementation. This occurrence has been previously reported in turbot (*Scophthalmus maximus* L.) skin mucus after bath administration of *W. cibaria* [31], and in rainbow trout after postbiotic administration [32]. Additionally, a down-regulation of the pro-inflammatory cytokines *TNF-α* and *IL-8* was reported at 72 h, and exclusively for *TNF-α* at 96 h.

These observations correspond with research like ours, where the treatment with postbiotics did not affect the expression of either of the mentioned cytokines in turbot and rainbow trout [31,32]. Additionally, the anti-inflammatory cytokine *IL-10* showed a tendency towards down-regulation at 72 and 96 h and on day 30 of the feeding trial, which is in line with the findings reported by Kahyani et al. [14] and Picchietti et al. [30] in rainbow trout and sea bass (*Dicentrarchus labrax*) after probiotic feed supplementation. The findings suggest that postbiotic supplementation may have immunomodulatory effects in fish. These interactions might be due to the presence of substances such as short-chain fatty acids, vitamins, peptidoglycan, or lipopolysaccharides within postbiotic products which can serve as stimulants of immune response in fish, as noted by Ang et al. [10] in their 2020 review. However, further research is necessary to understand the interaction between the postbiotic product and immune expression.

Previous studies have shown that dietary supplementation can improve the survival rate of rainbow trout against disease, including *Lactococcus garvieae* [33,34]. However, to the extent of current awareness, the results presented in this research are the first that have demonstrated the beneficial effect of postbiotic supplementation on rainbow trout survival upon *Y. ruckeri* infection. Postbiotics have been linked to promoting the proliferation and diversity of beneficial bacteria in the gastrointestinal tract, creating an unfavorable environment for the pathogens while simultaneously enhancing the overall health of the fish.

This perspective is consistent with current research on the higher prevalence of acid-lactic bacteria in the gut following 30 days of feed supplementation. However, the up-regulation of the *IL-1β* gene must also be considered, due to its role as a modulator of other antibacterial interleukin family members [35]. These facts, alongside the previously reported antibacterial activity of the postbiotics produced by *W. cibaria* on in vitro challenges against *Y. ruckeri*, might to some extent explain the mechanisms of the product in conferring protection against the pathogen [13].

## 5. Conclusions

This study demonstrated that a one-month supplementation of a postbiotic product derived from two strains of *W. cibaria* had a positive effect on the gut microbiota through increasing the concentration of acid-lactic bacteria. Additionally, the product up-regulated the cytokine *IL-1β* and improved survival rates in the face of *Y. ruckeri* infection. However, it did not affect animal growth. The new postbiotic product designed for rainbow trout has potential as a functional feed in aquaculture.

## Figures and Tables

**Figure 1 animals-14-00744-f001:**
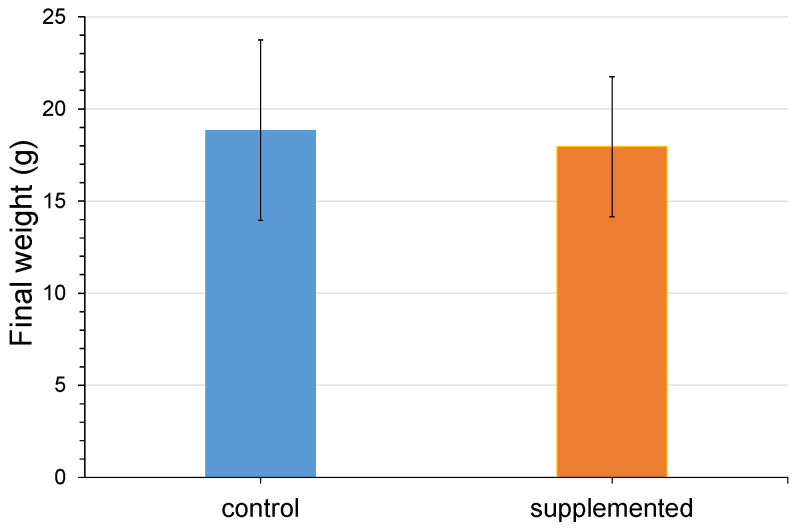
Rainbow trout weight after 30 days of feeding trial. Bars represent mean values of final weight error bars represent SD.

**Figure 2 animals-14-00744-f002:**
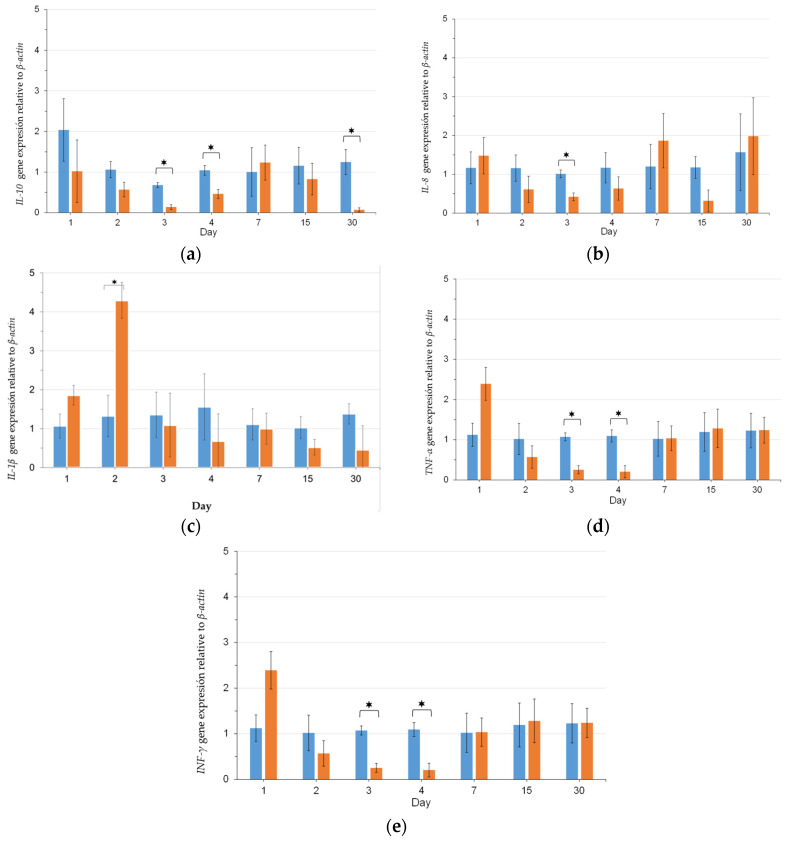
Relative gene expression (mean ± SD; *n* = 3) levels of cytokines in the head kidney of rainbow trout fed with control (■) and supplemented (■) diets. A bar with the asterisk (*) level indicates significant difference (*p* < 0.05); (**a**) relative gene expression of *IL-10*; (**b**) relative gene expression of *IL-8*; (**c**) relative gene expression of *IL-1β*; (**d**) relative gene expression of *TNF-α*; (**e**) relative gene expression of *INF-γ*.

**Figure 3 animals-14-00744-f003:**
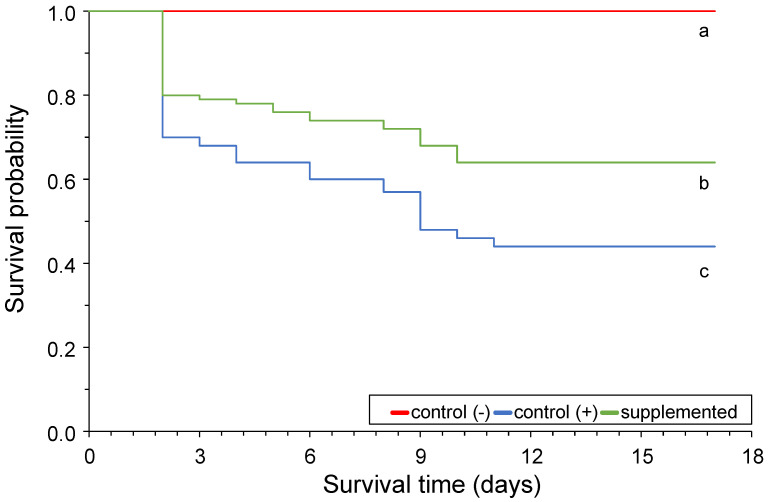
Survival curves of rainbow trout challenged with *Y. ruckeri* and treated with postbiotics. Survival for each group was calculated over the 17-day duration of the trial. Different letters indicate a significant difference (*p* < 0.05) between groups.

**Table 1 animals-14-00744-t001:** Primer used for Real-Time PCR.

Gene	GeneBank Accession Number	Product Size	Forward Primer	Reverse Primer
*β-actin*	NM001124235	186	GGACTGTTGACAGGAGATGG	ATGATGGAGTTGTAGGTGGTCT
*INF-γ*	AJ616215	151	GTGAGCAGAGGGTGTTGATG	GATGGTAATGAACTCGGACAG
*TNF-α*	XM020497470	125	GGCGAGCATACCACTCCTCT	TCGGACTCAGCATCACCGTA
*IL-8*	NM001140710	136	ATTGAGACGGAAAGCAGACG	CTTGCTCAGAGTGGCAATGA
*IL-10*	AB118099	70	CGACTTTAAATCTCCCATCGAC	GCATTGGACGATCTCTTTCTTC
*IL-1B*	AJ223954	91	ACATTGCCAACCTCATCATCG	TTGAGCAGGTCCTTGTCCTTG

The expression results were analyzed using the 2^−ΔΔCt^ method as described by Livak and Schmittgen [22]. The gene expression data were normalized to the reference gene β-actin.

**Table 2 animals-14-00744-t002:** Experimental conditions for real-time PCR reaction.

Process		Time	Temperature
Hot-start enzyme activation	1 cycle	3 min	95 °C
Amplification (44 cycles)	Denaturation	5 s	95 °C
	Annealing and extension	20 s	55 °C (fluorescence reading)
	Extension	5 s	79 °C (fluorescence reading)
Melting curve	Each 0.5 °C	every 5 s (fluorescence reading)	65–95 °C
End			10 °C

**Table 3 animals-14-00744-t003:** Means of total viable aerobic bacteria counts and acid-lactic bacteria counts in gut mucus samples of rainbow trout fed with diets supplemented with postbiotics and control diets.

		Counts (Log CFU/mL)	
	Day of Sampling	Control	Supplemented
Total aerobic bacteria	7	3.20 ± 0.08	3.03 ± 0.53
	15	5.03 ± 0.22	4.51 ± 0.36
	30	4.35 ± 0.41	4.63 ± 0.07
Acid-lactic bacteria	7	2.10 ± 0.14	2.36 ± 0.08
	15	2.26 ± 0.20	2.73 ± 0.52
	30	2.26 ± 0.20	3.42 ± 0.21 *

Results are expressed in means ± SD of three individual fish, each processed in duplicate (*n* = 3). Values in the same row showing * are significantly different (*p* < 0.05).

## Data Availability

Data are contained within the article.

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
