# Peer review of "A Novel Postbiotic Product Based on Weissella cibaria for Enhancing Disease Resistance in Rainbow Trout: Aquaculture Application"

_animals, 2024, doi:10.3390/ani14050744_

Round 1

Reviewer 1 Report

Comments and Suggestions for Authors

Considering that the authors have not given scientific and convincing answers to any of the questions raised in the first round of article review, unfortunately, I cannot approve the publication of the article.

Author Response

Reviewer 1.

Considering that the authors have not given scientific and convincing answers to any of the questions raised in the first round of article review, unfortunately, I cannot approve the publication of the article.

Dear Reviewer,

We would like to thank you for taking the time to review our manuscript and following up with questions about previous revisions.

Regarding the previous reviews, we must respectfully disagree with your observation of not providing a scientific basis to support our research during the given revision, since we carefully followed up with every concern raised during the initial revision process, making substantial changes throughout the entire manuscript. We understand that some of the arguments used to uphold the experimental design were orientated to the availability of the research facilities used for the experiments, as well as the limitations that the ethics committee might subject the animal experimentation protocols to. Nevertheless, to obtain reliable results, that at the same time would align with the previously cited limitations, a deep scientific literature review was made to support the design and to do so, the presented work was supported with various scientific references during the revision process, some of them even being cited in the manuscript, to grant access to the readers to other researchers where the numbers of replicates where equal to us and sufficient to obtain reliable results with similar size effects over survival after treatment.

We would like to include the lists of research used to support our design:

  1. Effect of a secondary metabolite with biological application on disease resistance of Atlantic salmon, executed with 2 group per treatment during challenge. They obtained significative results (p < 0.05) with an effect size of 20.6%.

Cortés H, Castillo-Ruiz M, Cañon-Jones H, Schlotterbeck T, San Martín R, Padilla L. (2023) In Vivo Efficacy of Purified Quillaja Saponin Extracts in Protecting against Piscirickettsia salmonis Infections in Atlantic Salmon (Salmo salar). Animals. 13(18):2845. https://doi.org/10.3390/ani13182845 

  1. Effect of a postbiotic product on disease resistance of rainbow trout, executed with 1 group per treatment during feeding trial and challenge. They obtained significative results (p < 0.05) with an effect size of 22.5%.

Mora-Sánchez, B., Balcázar, J. L., & Pérez-Sánchez, T. (2020). Effect of a novel postbiotic containing lactic acid bacteria on the intestinal microbiota and disease resistance of rainbow trout (Oncorhynchus mykiss). Biotechnology Letters42(10), 1957–1962. https://doi.org/10.1007/s10529-020-02919-9

  1. Effect of a prebiotic product on disease resistance of rainbow trout, executed with 1 group per treatment during feeding trial and challenge. They obtained significative results (p < 0.01) with an effect size of 45.0%.

Mora-Sánchez, B., Fuertes, H., Balcázar, J. L., & Pérez-Sánchez, T. (2020). Effect of a multi-citrus extract-based feed additive on the survival of rainbow trout (Oncorhynchus mykiss) following challenge with Lactococcus garvieae. Acta Veterinaria Scandinavica62(1). https://doi.org/10.1186/s13028-020-00536-0

  1. Effect of a postbiotic product on disease resistance of rainbow trout, executed with 1 group per treatment during feeding trial and challenge. They obtained significative results (p < 0.02) with an effect size of 14.7%.

Pérez-Sánchez, T., Mora-Sánchez, B., Vargas, A., & Balcázar, J. L. (2020). Changes in intestinal microbiota and disease resistance following dietary postbiotic supplementation in rainbow trout (Oncorhynchus mykiss). Microbial Pathogenesis142, 104060.  https://doi.org/10.1016/j.micpath.2020.104060

We hope that the presented arguments, the scientific research articles alongside the improvements made by following up on the recommendation of the other reviewers have improved the manuscript to meet your standards and the ones of the Journal Animals.

Reviewer 2 Report

Comments and Suggestions for Authors

The safety and the effect of the postbiotic product on growth performance, gut microbial communities, immuno-modulation, and protection against Yersinia ruckeri infection was evaluated. This research has certain application value, but there are still many shortcomings in the article.

1.      “Biosafety evaluation of postbiotic products”: The design of the experiment was not clearly described.

2.      Line 133: “animals were distributed in six groups of 9 individuals in tanks of 100 L”, What does that mean?

3.      Line134: The replication of the infection test is insufficient, and 3-4 replicates should be set.

4.      Line 137: “MSS 222” should be “MS 222”.

5.      Line 147: The 30-day breeding cycle is insufficient, and the feeding trail should last 56-90 days.

6.      Line 152-154, How many duplications are there for each group?

7.      Line 157, line 161, What is the basis for sampling at these time points?

8.      Line 233-235: Grouping is not scientific, at least 3-4 tanks per group.

9.      The amount of research data is insufficient, and growth and nutrition indicators should be supplemented.

Comments on the Quality of English Language

Quality of English Language is fine.

Author Response

Reviewer 2.

The safety and the effect of the postbiotic product on growth performance, gut microbial communities, immuno-modulation, and protection against Yersinia ruckeri infection was evaluated. This research has certain application value, but there are still many shortcomings in the article.

Dear reviewer,

The authors extend their sincere gratitude for your meticulous review of the manuscript. Every comment and suggestion you provided has been thoroughly considered, and we have endeavored to address each one individually. We have implemented improvements to the manuscript in alignment with your valuable insights. Furthermore, we have carefully incorporated the advice and recommendations from the entire reviewing team. We firmly believe that the comprehensive revisions made have significantly enhanced the quality of the manuscript.

  1. “Biosafety evaluation of postbiotic products”: The design of the experiment was not clearly described.

Thank you very much for your observation. We have made major changes to the section to better explain the design of the biosafety evaluation. You will find the improvements on lines 124-126; and 135-143.

  1. Line 133: “animals were distributed in six groups of 9 individuals in tanks of 100 L”, What does that mean?

Thank you for pointing this out. We have included a more exhaustive explanation to make it clearer to the reader. You will find this change on lines 136-142.

  1. Line134: The replication of the infection test is insufficient, and 3-4 replicates should be set.

Thank you for your suggestion. The design of the experiment was constructed by guidance of the research article published by Medina et al., 2020, where only 2 replicates per group were used during the biosafety evaluation of potential probiotic strains. To support the design, we have made the inclusion of the reference on line 126.

Medina, M.; Sotil, G.; Flores, V.; Fernández, C.; Sandoval, N. In vitro assessment of some probiotic properties and inhibitory activity against Yersinia ruckeri of bacteria isolated from rainbow trout Oncorhynchus mykiss (Walbaum). Aquac. Rep. 2020, 18, 100447.

  1. Line 137: “MSS 222” should be “MS 222”.

Thank you very much for pointing this out. It has been revised accordingly, and the entire manuscript has been checked out to spot any other similar misspells.

  1. Line 147: The 30-day breeding cycle is insufficient, and the feeding trail should last 56-90 days.

Thank you for your observation. The time of the feeding trial was established according to the research of Mora-Sanchez et al., 2020 and Sorroza et al., 2012 where the effect of biological tools administered through fed on disease resistance in fish was evaluated by supplementing the animals on a range of 25-30 days.

Mora-Sánchez, B., Balcázar, J. L., & Pérez-Sánchez, T. (2020). Effect of a novel postbiotic containing lactic acid bacteria on the intestinal microbiota and disease resistance of rainbow trout (Oncorhynchus mykiss). Biotechnology Letters42(10), 1957–1962. https://doi.org/10.1007/s10529-020-02919-9

Sorroza, L., Padilla, D., Acosta, F., Román, L., Grasso, V., Vega, J., Real, F., (2012). Characterization of the probiotic strain Vagococcus fluvialis in the protection of European sea bass (Dicentrarchus labrax) against vibriosis by Vibrio anguillarum. , 155(2-4), 369–373. doi:10.1016/j.vetmic.2011.09.013  

  1. Line 152-154, How many duplications are there for each group?

Thank you for your question. Since the main objective of our research was to determine the influence of the postbiotic product over microbiota, and immune system stimulation, to improve resistance to yersiniosis, no replicas were used during the feeding period, animals were divided into replicas during the challenge, following the methodology described by Sorroza et al., 2012.

Sorroza, L., Padilla, D., Acosta, F., Román, L., Grasso, V., Vega, J., Real, F., (2012). Characterization of the probiotic strain Vagococcus fluvialis in the protection of European sea bass (Dicentrarchus labrax) against vibriosis by Vibrio anguillarum. , 155(2-4), 369–373. doi:10.1016/j.vetmic.2011.09.013  

  1. Line 157, line 161, What is the basis for sampling at these time points?

Thank you for pointing this out. For both studies, we followed up with other scientific research testing products like the one we were investigating, this is why for microbiological observations, samplings were performed as described by Balcazar et al., 2007. On the other hand, we intended to determine when and for how long the immune stimulation will occur, for which we used the research conducted by Muñoz-Atienza et al., 2014 and Raja et al., 2022. as guidance.

Balcázar, José Luis; de Blas, Ignacio; Ruiz-Zarzuela, Imanol; Vendrell, Daniel; Calvo, Ana Cristina; Márquez, Isabel; Gironés, Olivia; Muzquiz, José Luis (2007). Changes in intestinal microbiota and humoral immune response following probiotic administration in brown trout (Salmo trutta). British Journal of Nutrition, 97(3) doi:10.1017/s0007114507432986  

Muñoz-Atienza E, Araújo C, Magadán S, Hernández PE, Herranz C, Santos Y, Cintas LM. In vitro and in vivo evaluation of lactic acid bacteria of aquatic origin as probiotics for turbot (Scophthalmus maximus L.) farming. Fish Shellfish Immunol. 2014 Dec;41(2):570-80. doi: 10.1016/j.fsi.2014.10.007

Khan, M. I. R., Kamilya, D., Choudhury, T. G., & Rathore, G. (2022). Dietary administration of a host-gut derived probiotic Bacillus amyloliquefaciens COFCAU_P1 modulates immune-biochemical response, immune-related gene expression, and resistance of Labeo rohita to Aeromonas hydrophila infection. Aquaculture, 546, 737390. doi:10.1016/j.aquaculture.2021.73

  1. Line 233-235: Grouping is not scientific, at least 3-4 tanks per group.

Thank you very much for your suggestion. We would like to explain further this matter, to fulfil the requirements of the Ethics Committee and the Animal Welfare Service of the Navarre Government on the reduction of animals used for experimentation, we considered that the design most adequate to follow was the one described by Mora-Sánchez, et al., 2020A, 2020B; and Pérez-Sánchez, et al., 2020., were significative results with a similar size effect were obtained by using only 1-2 replicas during challenge experiments when testing the effect of postbiotic and prebiotic products on rainbow trout.

Mora-Sánchez, B., Balcázar, J. L., & Pérez-Sánchez, T. (2020). Effect of a novel postbiotic containing lactic acid bacteria on the intestinal microbiota and disease resistance of rainbow trout (Oncorhynchus mykiss). Biotechnology Letters42(10), 1957–1962. https://doi.org/10.1007/s10529-020-02919-9

Mora-Sánchez, B., Fuertes, H., Balcázar, J. L., & Pérez-Sánchez, T. (2020). Effect of a multi-citrus extract-based feed additive on the survival of rainbow trout (Oncorhynchus mykiss) following challenge with Lactococcus garvieae. Acta Veterinaria Scandinavica62(1). https://doi.org/10.1186/s13028-020-00536-0

Pérez-Sánchez, T., Mora-Sánchez, B., Vargas, A., & Balcázar, J. L. (2020). Changes in intestinal microbiota and disease resistance following dietary postbiotic supplementation in rainbow trout (Oncorhynchus mykiss). Microbial Pathogenesis142, 104060.  https://doi.org/10.1016/j.micpath.2020.104060

  1. 9.      The amount of research data is insufficient, and growth and nutrition indicators should be supplemented.

Thank you very much for your suggestion. We would like to explain deeper on this matter, we understand that analyzing growth parameters and somatic indexes, as well as conversion ratios could be an interesting approach when testing novel feed additives, nevertheless, our primary objective in this investigation was to evaluate the effect over disease resistance in rainbow trout, particularly against Yersinia ruckeri. This is why we decided to follow Mora-Sánchez, et al., 2020 approach, focusing on other benefits that the postbiotics might have. Nevertheless, we would like to express our interest in this approach for future research on the product in development.

Mora-Sánchez, B., Fuertes, H., Balcázar, J. L., & Pérez-Sánchez, T. (2020). Effect of a multi-citrus extract-based feed additive on the survival of rainbow trout (Oncorhynchus mykiss) following challenge with Lactococcus garvieae. Acta Veterinaria Scandinavica62(1). https://doi.org/10.1186/s13028-020-00536-0

Reviewer 3 Report

Comments and Suggestions for Authors

The manuscript focuses on in vivo testing of the postbiotic product based on two strains of Weissella cibaria. I think it is a solid work. I have only minor comments, mainly to improve readability.

Title:

I would suggest adding Weissella cibaria to the title to make it more precise. For example: “A novel postbiotic product based on Weissella cibaria targeting disease resistance in rainbow trout”.

Abstract:

Lines 24-25: I would simplify the sentence to “… diet supplementation with 0.50% postbiotics during a 30-day feeding trial…”

Lines 26-27: I would move this sentence up, after the “… strains used in postbiotic production.”

Introduction:

Line 41: Delete “rearing” as it is superfluous.

Line 46: Delete “caused by”.

Lines 49-50: This sentence is repetitive, I would delete it.

Line 84: Use “test” instead of “substantiate”.

Line 91: Delete “to assess their safety”.

Line 92: Delete “… mean of the final weight of both the supplemented and non-supplemented groups.”

Materials and methods:

Line 99: “The bacteria were incubated” instead of “Incubating them”.

Line 102: “by centrifugation” instead of “by centrifuging them”.

Lines 104 and 105: I would use the term “administration” instead of “infection” when live cells of W. cibaria were used (since they are not pathogenic). Here and in the rest of the manuscript.

Line 137: MS 222 (not MSS 222)

Line 144-145: Please describe how this microbiological analysis was performed or add a suitable reference.

Results:

Section 3.2. I assume that no mortality was observed in either group during the feeding trial? This information can be added as the control fed with supplemented feed is missing from the Yersinia ruckeri infection trial.

Line 276: What do you mean by p < 0.07? This is not significant, and then the exact p-value should be given.

Table 3. These results should be presented in a graph for easier understanding.

Figure 2e. Please correct the title of y axis to TNF.

Discussion:

Each time the authors refer to postbiotic products, I suggest specifying the type or origin of the product, as postbiotics are highly variable.

Lines 334-335: I would change the sentence to “This is of importance since disease outbreaks…”

Lines 336-340: I don’t understand this part. What do the authors mean by “for which the safety evaluations of the candidates have been firmly recommended”? Disease outbreaks are mentioned…

Lines 349-353: I would delete this part. There is no need to discuss it, as this was not the result of this study.

Comments on the Quality of English Language

Although I am not a native speaker, I think the language can be improved. For example, use the terms “lactic acid bacteria” and “significant” throughout the manuscript, instead of (“Acid-lactic” and “significative”.

Line 28: “treated”, not “traded”

Line 45: spp. should not be italicized

Line 204: “reverse” instead of “revers”

Line 213: “in Table 2” (not on)

Table 1: “actin”, not “actine”

Table 2: “Hot start”, not “Host start”, and “denaturation”, not “denaturalization”

Line 343: Replace “over” with “on the”

Line 344: Replace “which” with “in”

Reviewer 4 Report

Comments and Suggestions for Authors

The paper “A Novel Postbiotic Product Targeting Disease Resistance in Rainbow Trout”

Describes an interesting and well executed experiment aimed to test the potential value of a postbiotic obtained from two specific bacterial strains of Weisella cibaria to enhance the immune status of rainbow trout and to prevent infection by Yersinia ruckeri. The study comprises a preliminary evaluation of the safety of the postbiotic preparation followed by an evaluation of the effect its oral administration on growth and of some functional indicators selected to evidence potential changes either in the profile of intestinal microbiota or the immune status (gene expression of cytokines and interleukins). The final part presents results of a challenge which demonstrates a significant positive effect of postbiotic consumption against induced infection by Y. ruckeri

Only some comments:

L88-92; Change the order of objectives to adapt better to the experimental sequence: 1) biosafety evaluation 2) results of feeding trial either on growth performance intestinal microbiota or immune status and 3) Infection challenge

L98 > Please indicate in more detail the procedure used for isolation

Lane 109 > Any information on the main components present in the postbiotic obtained from the bacterial cultures?

L121> The link does not provide direct information to diet composition

L154> Why this dose? Please justify the scientific basis for this.

L167> No description of the procedure used to evaluate fish growth in the experiment

L248> Please indicate how many fish were inoculated in each treatment

L280> In Fig 1 indicate what represents error bars; SME, SD?

L287> In table 3 better include the sentence “Values in the same row showing * are significantly different with p< 0.05”

Round 2

Reviewer 1 Report

Comments and Suggestions for Authors

I read the article once again and unfortunately, the changes presented and the answers of the authors were not convincing for me, and therefore I cannot recommend this article for publication.

Author Response

Dear Reviewer,

The authors sincerely appreciate your time and effort in reviewing the manuscript. However, they respectfully disagree with your assessment. The manuscript has been substantially improved during the review process. All the comments from the three reviewers have been addressed. Extensive explanations were also provided,

The authors have carefully reviewed all methodological and experimental design aspects. They have relied on existing research. The introduction, results, and discussion sections have been modified according to the reviewers' recommendations. In addition, they have proofread the text to fix both stylistic and typographical errors.

The authors believe that their research has contributed relevant knowledge in the fields of postbiotics, aquaculture, and biological tools. For all these reasons, they consider the results to be of enough quality for publication,

Yours sincerely